# Frequency-Dependent Antioxidant Responses in HT-1080 Human Fibrosarcoma Cells Exposed to Weak Radio Frequency Fields

**DOI:** 10.3390/antiox13101237

**Published:** 2024-10-15

**Authors:** Hakki Gurhan, Frank Barnes

**Affiliations:** Department of Electrical, Computer and Energy Engineering, University of Colorado Boulder, 1111 Engineering Dr 425 UCB, Boulder, CO 80309, USA

**Keywords:** radio frequency fields, cancer cells, oxidative stress, mitochondrial superoxide, reactive oxygen species, antioxidant enzymes, superoxide dismutase, reduced glutathione, peroxidase, radical pair mechanism

## Abstract

This study explores the complex relationship between radio frequency (RF) exposure and cancer cells, focusing on the HT-1080 human fibrosarcoma cell line. We investigated the modulation of reactive oxygen species (ROS) and key antioxidant enzymes, including superoxide dismutase (SOD), peroxidase, and glutathione (GSH), as well as mitochondrial superoxide levels and cell viability. Exposure to RF fields in the 2–5 MHz range at very weak intensities (20 nT) over 4 days resulted in distinct, frequency-specific cellular effects. Significant increases in SOD and GSH levels were observed at 4 and 4.5 MHz, accompanied by reduced mitochondrial superoxide levels and enhanced cell viability, suggesting improved mitochondrial function. In contrast, lower frequencies like 2.5 MHz induced oxidative stress, evidenced by GSH depletion and increased mitochondrial superoxide levels. The findings demonstrate that cancer cells exhibit frequency-specific sensitivity to RF fields even at intensities significantly below current safety standards, highlighting the need to reassess exposure limits. Additionally, our analysis of the radical pair mechanism (RPM) offers deeper insight into RF-induced cellular responses. The modulation of ROS and antioxidant enzyme activities is significant for cancer treatment and has broader implications for age-related diseases, where oxidative stress is a central factor in cellular degeneration. The findings propose that RF fields may serve as a therapeutic tool to selectively modulate oxidative stress and mitochondrial function in cancer cells, with antioxidants playing a key role in mitigating potential adverse effects.

## 1. Introduction

Cancer remains one of the leading causes of mortality worldwide, necessitating the ongoing exploration of advanced therapeutic and preventive strategies. A growing area of interest is the impact of radio frequency (RF) exposure, which has become ubiquitous in modern life [1,2]. Since the 1920s, RF exposure has significantly increased due to the widespread use of various communication and security devices. These electromagnetic fields (EMFs) interact with biological systems, which contain charged ions and polarized molecules, impacting cell membranes, transmembrane potentials, and cell cycles [3]. Recent studies in quantum biology have extended this understanding, suggesting that weak magnetic fields, including those produced by RF exposure, can modulate biological processes at the quantum level [4,5]. This modulation occurs particularly through mechanisms involving spin states and radical pair recombination. [6,7]. Such quantum effects may play a significant role in how RF fields affect cellular processes, including the generation of oxidative stress and reactive oxygen species (ROS) [8].

At high concentrations, free radicals and radical-derived nonradical reactive species are hazardous to living organisms and can damage all major cellular constituents [9]. However, at lower concentrations, nitric oxide (NO), superoxide anion (O_2_•^−^), and related ROS play important roles as regulatory mediators in mammalian cell signaling processes. For instance, intracellular concentrations of hydrogen peroxide (H_2_O_2_) in non-stressed mammalian cells are typically in the range of 1–100 nM, while levels can rise to several micromolar under oxidative stress conditions. Similarly, O_2_•^−^ levels in mitochondria are generally kept low through efficient dismutation by superoxide dismutase (SOD) to produce H_2_O_2_. The physiological concentration of NO ranges from 20 nM to 1 µM, depending on the cell type and the extent of stimulation of nitric oxide synthase (NOS). These moderate levels are crucial for their function as signaling molecules, enabling the regulation of vascular tone, immune response, and apoptosis, among other processes [10,11]. The electron transport chain (ETC) in mitochondria is a critical site for ROS production, primarily through the activity of various redox-active molecules that exhibit hyperfine frequencies [12].

Barnes and Freeman emphasized that biological systems respond to weak EMFs at energy levels well below current safety guidelines, indicating a need for revised exposure standards [13]. For example, magnetic fields around 35 µT, similar to the geomagnetic field, have been shown to affect human brainwave activity, demonstrating the brain’s sensitivity even at low intensities [14]. Static magnetic fields (SMFs) of 300 and 400 µT have been observed to accelerate HT-1080 fibrosarcoma cancer cell growth and modify ROS concentrations, while 0.5 µT and 600 µT fields inhibit growth [15]. Usselman et al. found that RF magnetic fields of 20 µT rms at 1.4 MHz combined with a 50 µT static field reduced O_2_•^−^ levels by 36% and H_2_O_2_ by 21% in the perpendicular orientation, with different effects in the parallel orientation [16]. These findings underscore the need for updated EMF exposure guidelines.

The radical pair mechanism (RPM) explains how weak magnetic fields can influence chemical reactions in biological systems. RPM involves pairs of molecules or atoms with unpaired electrons, which can interact with each other and magnetic fields, affecting their recombination rates and pathways [17,18]. According to Grissom, magnetic fields can modulate the rate of radical pair recombination in biological systems through several mechanisms, including hyperfine-induced intersystem crossing and spin-orbit coupling [19]. These mechanisms can significantly alter the production and recombination rates of radical pairs, thus affecting ROS generation and the oxidative stress response in cells.

Sheppard et al. noted that vibrational modes below the far-infrared range are dampened by water, highlighting RPM and dielectric heating as key interactions for RF fields at common exposure levels [20]. Timmel and Hore showed that oscillating magnetic fields, when matching hyperfine splitting frequencies, can change radical pair reaction yields by up to 25%, emphasizing singlet-triplet interconversion [21]. Austvold et al. demonstrated how magnetic fields modulate ROS partitioning in flavoenzymes, which helps explain how RF fields may influence ROS production and antioxidant enzyme activity in cancer cells [22]. Zadeh-Haghighi and Simon et al. highlighted similar findings, demonstrating that weak magnetic fields could modulate radical pair reactions, potentially impacting ROS dynamics in cancer cells [23].

### 1.1. Oxidative Stress and Cancer

Oxidative stress is a critical factor in cancer progression, primarily through the generation of ROS that leads to DNA damage, lipid peroxidation, and protein modifications [24]. These oxidative modifications can promote tumor initiation, progression, and resistance to therapy. These processes are not only pivotal in cancer but are also key contributors to aging and other diseases [25]. The mitochondria, as major producers of ROS, play a pivotal role in these processes [26]. H_2_O_2_, a critical ROS, is particularly important in cancer cells, where its regulated production and detoxification are crucial for tumor cell survival and progression [27]. ROS, such as superoxide and H_2_O_2_, are primarily generated within the mitochondria during cellular respiration. While Complexes I and III of the ETC are the primary sources, Complex II also contributes under certain conditions, with electron leakage leading to ROS formation during ATP production [28]. When ROS are produced in excess, they overwhelm the cellular antioxidant defense mechanisms, leading to oxidative stress [29]. Each ROS type has distinct cellular targets and effects. For example, H_2_O_2_ plays a pivotal role in cell signaling by selectively modifying protein functions. However, improper regulation of H_2_O_2_ can lead to the generation of highly reactive hydroxyl radicals (•OH), causing extensive damage to cellular components, such as DNA, proteins, and lipids [30].

Sies et al. emphasized the importance of differentiating between oxidative eustress, which involves physiological deviations from the steady-state redox set point that aid in adaptive signaling, and oxidative distress, which results from excessive oxidative challenge and causes biomolecular damage and disrupted signaling pathways [31]. In cancer cells, the balance between ROS production and antioxidant defenses is often disrupted [32]. This imbalance can enhance cancer cell survival, proliferation, and metastasis by activating various signaling pathways [33].

### 1.2. Antioxidants Defense Mechanisms

Cells possess a complex antioxidant defense system to mitigate oxidative damage. Key antioxidant enzymes include SOD, which converts superoxide into H_2_O_2_; catalase (CAT), which breaks down H_2_O_2_ into water and oxygen; and glutathione peroxidase (GPx), which reduces H_2_O_2_ and lipid peroxides using glutathione (GSH). These enzymes are vital not only for preventing oxidative damage but also for regulating ROS levels involved in signaling pathways [34]. As shown in Figure 1, the activities of these antioxidant enzymes are interconnected and crucial in mitigating oxidative stress.

SODs, a group of critical antioxidant enzymes, catalyze the dismutation of superoxide radicals into H_2_O_2_. This reaction forms the first line of defense against ROS. The H_2_O_2_ produced can react with nearby reduced thiols or transition metals, exacerbating oxidative stress if not adequately detoxified by subsequent enzymatic action. The conversion of H_2_O_2_ to water is crucial to prevent oxidative damage, facilitated by enzymes like catalase and GPx, which use GSH to reduce H_2_O_2_. The reduction of GSH to GSSG (glutathione disulfide) is a vital marker of oxidative stress; a high GSH/GSSG ratio is typically indicative of a reduced, non-stressed state, while a low ratio points to oxidative imbalance.

### 1.3. Radical Pair Mechanism

Radical pairs are formed through normal metabolic processes, such as mitochondrial respiration. These pairs consist of molecules or atoms with unpaired electrons, whose interactions can be influenced by magnetic fields. According to Woodward et al. [36], the RPM explains how magnetic fields alter chemical yields by influencing the spin states of radical pairs. Brocklehurst [37] describes how the transition between singlet and triplet states of radical pairs is modulated by Zeeman and hyperfine interactions, affecting reaction kinetics and product formation.

Hyperfine interactions occur between unpaired electrons and nuclear spins of neighboring atoms, modulating the reactivity of photoinduced radicals like those involving flavin adenine dinucleotide (FAD) and tryptophan. Zeeman interactions, which are caused by external magnetic fields, influence singlet-triplet transitions and the efficiency of ROS production, making the RPM a potential explanation for magnetic field effects on biological systems. Barnes and Greenebaum [38] propose that weak magnetic fields, such as those used in our study, may modify radical pair recombination rates, thus altering ROS concentrations and subsequent oxidative stress responses.

In our study, the RPM provides a plausible explanation for how RF fields influence oxidative stress and antioxidant enzyme activities. RF exposure at specific frequencies may interact with hyperfine couplings within radical pairs, affecting ROS production, as explained in the context of magnetic field effects on biological systems [39]. The objective of this study is twofold: first, to investigate how RF exposure modulates oxidative stress in cancer cells, potentially contributing to oncogenesis; and second, to explore whether specific frequencies can sensitize cancer cells to therapeutic interventions. Identifying the optimal frequency that maximizes or minimizes biological effects, such as ROS generation or antioxidant activity, could be crucial for developing new therapeutic strategies. Further studies will also be needed to assess the effects of RF on normal cells to evaluate potential off-target effects.

### 1.4. Influence of Hyperfine Resonances on Electron Transport Chain (ETC)

Enhancing mitochondrial function and antioxidant defenses offers potential for cancer prevention and treatment of age-related diseases [40]. The ETC in mitochondria is a major site of ROS production, primarily due to the activity of redox-active molecules like FAD, ubiquinone (CoQ10), and iron-sulfur clusters. These molecules often form radical pairs sensitive to external magnetic fields due to hyperfine interactions. Maly et al. [41] highlight the crucial role of Fe-S clusters, particularly cluster N2 in mitochondrial Complex I, in modulating electron transfer and ROS production.

Hyperfine interactions within iron-sulfur clusters, particularly in Complex I, play a significant role in electron transfer processes. These interactions modulate the singlet-triplet conversion rates of radical pairs, thereby affecting the efficiency of ROS production. Ubiquinone, an essential electron carrier, forms semiquinone radicals during its redox cycle. The hyperfine interactions within these radicals can influence their recombination rates, impacting overall electron flow through the ETC and contributing to oxidative stress. Additionally, the Fe-S clusters in Complex I, II, and III, due to their hyperfine interactions, are susceptible to alterations in redox states, which can lead to increased ROS production under certain RF field conditions. As shown in Figure 2, the presence of prominent peaks in the ESEEM spectra indicates significant hyperfine couplings, which can influence the efficiency of electron transfer and ROS production.

### 1.5. Radio Frequency (RF) Field Effect on Antioxidants

The interconnected activities of SOD and GPx illustrate a coordinated defense against oxidative stress and play a pivotal role in maintaining redox homeostasis [42]. SOD mitigates the initial superoxide radicals, converting them to H_2_O_2_, which GPx then reduces to water. Any imbalance in this system can lead to the accumulation of ROS, subsequent oxidative stress, and improper signaling through various cellular pathways [43]. Non-ionizing radiation, like RF fields, alters the spin states of radical pairs, influencing ROS production through spin-state modulation [44]. For example, if RF exposure increases ROS production beyond the capacity of SOD and GPx to manage, this could result in increased oxidative stress, as indicated by changes in the GSH/GSSG ratio. This disruption in the redox balance has been implicated in the progression of various diseases, including cancer. Furthermore, the influence of RF fields on radical pairs, especially those formed during metabolic processes, could alter ROS production and antioxidant enzyme activities.

To explore the influence of RF exposure on these key antioxidant defenses, a detailed examination of the mechanisms by which RF fields affect cellular oxidative stress is necessary. One plausible mechanism involves the application of RF fields at frequencies near the hyperfine resonances of key ETC molecules such as FAD, CoQ10, and iron-sulfur clusters. These hyperfine interactions can alter the spin states and recombination rates of radical pairs formed during metabolic processes. The modulation of these spin states can lead to changes in mitochondrial function and overall oxidative stress levels.

The choice of the 2–5 MHz range is based on known hyperfine resonances of iron-sulfur clusters and other key components of the ETC within mitochondria. Studies have shown that certain iron-sulfur clusters in mitochondrial complexes, particularly Complex I, exhibit hyperfine transitions within this frequency range. By targeting these frequencies, the goal is to modulate electron transfer processes within the ETC, potentially affecting ROS production and antioxidant enzyme activity. If these changes result in increased ROS levels, the antioxidant enzymes SOD and GPx may become overwhelmed, failing to maintain redox homeostasis. Consequently, this would manifest as an altered GSH/GSSG ratio, indicating heightened oxidative stress. Understanding these interactions provides crucial insights into how RF fields modulate oxidative stress and highlights the potential for antioxidants to mitigate these effects.

Previous research [45] has explored the effects of RF exposure on oxidative stress and ROS levels in various cell types. Some studies have reported increased ROS production and oxidative damage in RF-exposed cells, while others have found no significant effects. This variability in findings underscores the complexity of RF effects, possibly due to differences in experimental conditions, cell types, and exposure parameters. This discrepancy highlights the need for further research to clarify the impact of RF exposure on oxidative stress and cancer progression.

## 2. Materials and Methods

### 2.1. Cell Culture

The HT-1080 human fibrosarcoma cell line (ATCC CCL-121, Manassas, VA, USA) was utilized in this study. Cells were maintained in Eagle’s Minimum Essential Medium (EMEM, ATCC 30-2003, Manassas, VA, USA) supplemented with 10% fetal bovine serum (FBS, ATCC 30-2020, Manassas, VA, USA). Cells were expanded in treated 75 cm^2^ flasks until they reached 70–90% confluence. Subsequently, the cells were transferred to 25 cm^2^ flasks or 24-well culture plates based on the experimental requirements.

The HT-1080 cell line was selected due to its rapid growth, which allowed the experiments to be conducted efficiently by reducing exposure time. The cells’ resilience to environmental stress and contamination ensured reliable results over extended experiments. Their hyperpolarized membrane potential also makes them particularly sensitive to RF fields, enabling clear observation of oxidative stress and antioxidant responses. Moreover, HT-1080 cells are cost-effective, with lower medium and material costs compared to normal cell lines, making them ideal for long-term studies. This study aims to identify specific frequencies that improve cell viability and antioxidant responses, potentially informing therapeutic strategies. Since HT-1080 is a cancer cell line and cancer is recognized as an aging-related disease, these findings are particularly relevant in the context of oxidative stress and aging.

### 2.2. Exposure System

The experiments were conducted using a dual chamber incubator (Model 3326, Forma Scientific Inc., Marietta, OH, USA) equipped with a mu-metal shielding box (MuShield Company, S7444, Londonderry, NH, USA) and Helmholtz coils to ensure precise control over magnetic field conditions. The mu-metal shielding box made from alloy 4 with a thickness of 0.125 inches, placed inside the incubator, effectively minimized background magnetic fields, allowing for the controlled application of SMFs and RF fields. Environmental parameters such as temperature, humidity, and CO_2_ levels were meticulously regulated to maintain consistent experimental conditions.

Following the protocol by Vuckovic et al. [46], the MF exposure apparatus was constructed and calibrated using commercially available materials. This setup enabled reliable and reproducible investigation of MF effects on biological samples. Uniform magnetic fields were generated using square Helmholtz coils with 30 turns each. One set of Helmholtz coils was used for the exposed cultures, while another served as the control. Control cells were maintained under an SMF of 45 µT, which approximates the Earth’s magnetic field in Boulder, Colorado, ensuring that the cells were exposed to a natural magnetic environment. Treated cells were subjected to both an SMF of 45 µT and varying RF fields. Both the SMF and RF magnetic fields were oriented vertically relative to the culture flasks or well plates. The coils were wound with 1.63 mm wire to address heating issues and maintain a stable temperature close to 37 °C. The dimensions of the coils (side = 15 cm) were selected to accommodate the culture flasks and well plates adequately. Figure 3 shows the experimental setup, including a detailed configuration of Helmholtz coils within the mu-metal cage.

The SMF was produced using a Triple Output DC Power Supply (Model E3631A, Agilent Technologies, Santa Clara, CA, USA), and measurements were taken with a fluxgate magnetometer (Model FGM-4D2N, Walker Scientific Inc., Worcester, MA, USA) with a resolution of 0.1 µT. The homogeneity of the exposure fields varied by ±5% across the region where culture flasks and well plates were placed. An EMF ranging from 2 MHz to 5 MHz was generated by a 15 MHz Function/Arbitrary Waveform Generator (Model 33120A, Agilent Technologies, Santa Clara, CA, USA). The RF magnetic field component, measured with a passive loop antenna (Model 100C, Beehive Electronics, Sebastopol, CA, USA), had a calibrated sensitivity of up to 3 GHz.

During RF field production, the amplitude of the voltage was adjusted to maintain constant flux density at different frequencies, considering the non-linear relationship between voltage, frequency, and magnetic field strength. The applied RF magnetic fields had an amplitude of 20 nT, with thermal effects deemed insignificant as temperature increases were less than 0.2 °C. Air circulation within the incubator was facilitated by a fan and eight 26 mm diameter holes in the mu-metal box, ensuring uniform temperature.

### 2.3. Experimental Procedure and Treatment Duration

Cells were treated with varying frequencies of RF fields for a total duration of 4 days. Each frequency-specific exposure was conducted under the controlled conditions detailed in the exposure system section. Control groups were maintained under an SMF of 45 µT without RF exposure for the same duration. The medium was refreshed on the second and third days to counteract the effects of prolonged culture, including nutrient depletion and metabolic waste accumulation. During the treatment period, cells were continuously monitored to ensure stable environmental conditions and consistent exposure. At the end of the 4-day treatment period, cells were harvested and prepared for subsequent assays to measure oxidative stress markers and antioxidant responses.

### 2.4. Oxidative Stress and Antioxidant Assays

#### 2.4.1. SOD Activity Assay

SOD activity in cell lysates was measured using the Superoxide Dismutase Colorimetric Activity Kit (Cat. No. EIASODC, Thermo Fisher Scientific, Waltham, MA, USA), following the manufacturer’s protocol. After treatment with RF fields ranging from 2 MHz to 5 MHz for 4 days, cells were harvested by centrifugation at 250× *g* for 10 min at 4 °C to form a pellet. The pellet was then resuspended in ice-cold Phosphate-Buffered Saline (PBS), and we homogenized the cells using the Mini-BeadBeater 16 (BioSpec Products, Cat. No. 607, Bartlesville, OK, USA) for 30 s with 0.5 mm glass disrupter beads (Cat. No. CLS-1835-BG5, Chemglass Life Sciences, Vineland, NJ, USA) to ensure complete cell lysis. The lysate was centrifuged at 1500× *g* for 10 min at 4 °C to remove debris, and the supernatant was collected for the SOD assay.

For the assay, 10 µL of each standard and sample was added to the wells of a clear 96-well plate. To each well, 50 µL of 1X Substrate solution was added, followed by the addition of 25 µL of 1X Xanthine Oxidase. The plate was incubated at room temperature for 20 min to allow the enzymatic reaction to occur. After the incubation period, the absorbance was measured at 450 nm using a microplate reader.

A standard curve was generated using serial dilutions of the provided SOD standard, ranging from 0 to 4 U/mL. The SOD activity in the cell lysate samples was determined by comparing their absorbance values to the standard curve. The activity was calculated by multiplying the value obtained from the standard curve by the appropriate dilution factor of the samples. This method allowed for the accurate quantification of SOD activity, reflecting the enzyme’s efficiency in scavenging superoxide radicals in the cell lysates.

#### 2.4.2. Peroxidase Assay

For reagent preparation, the Amplex^®^ Red reagent (Thermo Fisher Scientific, Waltham, MA, USA) was dissolved in 60 µL of dimethyl sulfoxide (DMSO) to prepare a 10 mM stock solution, which was used on the same day of preparation. The 5X Reaction Buffer was diluted to 1X with deionized water to achieve a final concentration of 0.25 M sodium phosphate, pH 7.4. The horseradish peroxidase (HRP) stock solution was prepared by dissolving the provided HRP in 1X Reaction Buffer to a concentration of 10 U/mL. The H_2_O_2_ working solution was diluted to 20 mM in 1X Reaction Buffer from a ~3% H_2_O_2_ stock solution. A working solution containing 100 µM Amplex^®^ Red reagent and 2 mM H_2_O_2_ was prepared by mixing 50 µL of 10 mM Amplex^®^ Red reagent stock solution, 500 µL of 20 mM H_2_O_2_ working solution, and 4.45 mL of 1X Reaction Buffer. To initiate the reaction, 50 µL of the Amplex^®^ Red reagent/H_2_O_2_ working solution was added to each well containing standards, controls, and samples.

After treatment with RF fields ranging from 2 MHz to 5 MHz for 4 days, cells were harvested and the reaction mixtures were incubated at room temperature for 30 min, protected from light. Fluorescence was then measured using a microplate reader with excitation at 530 nm and emission detection at 590 nm. Fluorescence readings were corrected for background by subtracting the values from the no-HRP control wells. This protocol allowed for the sensitive detection of peroxidase activity in the samples, with the fluorescent signal directly proportional to the amount of peroxidase present.

#### 2.4.3. Measurement of GSH

After treatment with RF fields ranging from 2 MHz to 5 MHz for 4 days, cells were lysed using Mammalian Cell Lysis Buffer 5X (Cat. No. ab179835, Abcam, Cambridge, UK) according to the manufacturer’s instructions. Briefly, adherent cells were grown to approximately 80% confluence, washed with PBS, and lysed with 1X Lysis Buffer. The lysates were then centrifuged at 1500 rpm for 5 min at room temperature, and the supernatant was collected and kept on ice.

Deproteinization of cell lysates was performed using the Deproteinizing Sample Preparation Kit—TCA (Cat. No. ab204708, Abcam, Cambridge, UK). For samples with protein concentrations less than 25 mg/mL, 150 µL of the sample was mixed with 15 µL of cold Trichloroacetic Acid Solution (TCA) and kept on ice for 15 min. The mixture was then centrifuged at 12,000× g for 5 min, and the supernatant was carefully collected. To neutralize the excess TCA, 10 µL of cold Neutralization Buffer was added to the supernatant and incubated on ice for 5 min.

The GSH/GSSG Ratio Detection Assay Kit II (Cat. No. ab205811, Abcam, Cambridge, UK) was used to measure the GSH. The procedure included the preparation of GSH standards, followed by the addition of deproteinized samples to a 96-well plate along with the standards. GSH Assay Mixture (GAM) was prepared according to the kit instructions. For GSH detection, 50 µL of GAM was added to the wells. The plate was then incubated at room temperature for 30 min, protected from light. Fluorescence was monitored at an excitation/emission wavelength of 490/520 nm using a fluorescence microplate reader.

#### 2.4.4. Measurement of Superoxide

Mitochondrial Superoxide levels were measured using MitoSOX™ Red mitochondrial superoxide indicator (Cat. No. M36008, Thermo Fisher Scientific, Waltham, MA, USA). HT-1080 cells were seeded at a density of 5000 cells per cm^2^ in the black 24-well plates with a flat and clear bottom (Cat. No. 82426, Ibidi, Gräfelfing, Germany), providing a growth area of 1.54 cm^2^ per well. Four wells in the middle of the plate were used for the experiment, and four points within each well were selected for measurements. The MitoSOX™ Red reagent stock solution was prepared by dissolving 50 μg of MitoSOX™ Red mitochondrial superoxide indicator in 13 μL of dimethylsulfoxide (DMSO) to make a 5 mM stock solution. The working solution was prepared by diluting this stock solution to 5 μM in PBS. Cells were incubated with 1 mL of the 5 μM MitoSOX™ reagent solution per well at 37 °C for 10 min, protected from light. After incubation, the cells were washed gently three times with a warm buffer to remove excess reagent. MitoSOX™ Red selectively detects mitochondrial superoxide and becomes highly fluorescent upon oxidation by O_2_•^−^. Fluorescence measurements were taken using an appropriate fluorescence microscope at excitation/emission wavelengths of 500/582 nm.

Extracellular superoxide levels were measured using Electron Paramagnetic Resonance (EPR) with an X-band spectrometer (Model EMXnano, Bruker Corporation, Billerica, MA, USA) at 77 K, following the described protocol. HT-1080 cells were cultured in 6-Well Tissue Culture-Treated Multiple Well Plates (Polystyrene) (Cat. No. 229106, CellTreat, Pepperell, MA, USA), which have a surface area of 3.85 cm^2^ per well. Cells were seeded at a density of 5000 cells per cm^2^. On the day of the experiment, 15 μL of the SOD 30 KU stock was mixed with 435 μL of PBS to create a SOD 1 KU working solution. Then, 9 μL of this solution was added to the wells designated for CMH + SOD treatment, while 9 μL of PBS was added to the control wells designated for CMH treatment. CMH refers to 1-Hydroxy-3-methoxycarbonyl-2,2,5,5-tetramethylpyrrolidine, a spin probe used in EPR to detect superoxide radicals.

The cell media were aspirated, and the cells were washed with 1 mL of Krebs-Henseleit Buffer (KHB), a physiological buffer solution. Then, 300 μL of KHB with 100 μM DTPA was added to each well. DTPA (Diethylenetriaminepentaacetic acid) is a chelating agent that binds metal ions, which can interfere with the detection of superoxide radicals by forming metal-superoxide complexes. In this context, DTPA ensures the accurate measurement of extracellular superoxide levels by preventing metal ion interference. Subsequently, 9 μL of the SOD working solution was added to the wells designated as CMH + SOD, and 9 μL of PBS was added to the control wells (CMH). The cells were incubated for 10 min at 37 °C and 5% CO_2_. After the incubation, 8 μL of CMH (10 mM stock) was added to all wells.

To ensure accurate measurement of extracellular superoxide levels, a background signal was prepared by adding 300 μL of KHB with DTPA and 8 μL of CMH (10 mM stock) to a 1.5 mL Eppendorf tube. Both the experimental plate and the Eppendorf tube were incubated for 50 min at 37 °C and 5% CO_2_. After incubation, they were immediately placed on ice. The buffer from each well was collected and transferred into labeled 1.5 mL Eppendorf tubes, kept on ice. A volume of 150 μL of the buffer was pipetted into PTFE tubing and sealed, and the same procedure was followed for the background tube. Samples were flash-frozen in liquid nitrogen, transferred to cryopreservation tubes, and placed in the Finger Dewar for data acquisition. EPR measurements were then performed at 77 K. Cell confluence was used for normalization to ensure consistency in the measurements.

#### 2.4.5. Cell Viability

Cell viability was assessed using the CellTiter 96^®^ Non-Radioactive Cell Proliferation Assay (Cat. No. G4000, Promega, Madison, WI, USA), following the manufacturer’s protocol. HT-1080 cells were cultured in 48-Well Tissue Culture-Treated Polystyrene Plates (Cat. No. 229148, Corning, Corning, NY, USA). In each experiment, 14 wells in the middle of the plate were seeded with cells at a density of 5000 cells per cm^2^, corresponding to 4200 cells per well (each well has a surface area of 0.84 cm^2^).

Both control and treated cells were exposed to experimental conditions for 4 days. After the RF treatment period, the metabolic activity of the cells, serving as an indicator of viability, was determined by adding 30 µL of Dye Solution to each well, followed by incubation at 37 °C for 3 h. After the incubation, 200 µL of Solubilization/Stop Solution was added to each well to dissolve the formazan product. The absorbance was recorded at 570 nm using a multimode microplate reader (Model Varioskan™ LUX, Thermo Fisher Scientific, Waltham, MA, USA), with a reference wavelength of 630 nm, 1 h after the addition of the Solubilization/Stop Solution.

### 2.5. Statistical Analysis

The statistical analyses were performed using Origin Pro 2023 software (OriginLab, Northampton, MA, USA). Data were evaluated using one-way ANOVA followed by Student’s *t*-test for post hoc comparisons. Results are expressed as mean ± standard deviation (SD). The total number of samples is indicated by ‘*n*’, while ‘*N*’ represents the number of independent experimental replications. Statistical significance was assigned at three levels: * *p* < 0.05, ** *p* < 0.01, and *** *p* < 0.001. For clarity, data are presented with negative control values (untreated cells) normalized to 1.

## 3. Results

### 3.1. Antioxidant Enzyme Activities: SOD

SOD is a crucial antioxidant enzyme that scavenges superoxide radicals by converting them into H_2_O_2_, thus maintaining cellular redox homeostasis and preventing oxidative damage. Total cellular SOD activity was measured to assess the antioxidant capacity necessary for redox balance under normal conditions. A balance between antioxidants and oxidants prevents oxidative damage, with ROS like superoxide, hydroxyl radicals, and H_2_O_2_ being natural byproducts of metabolism. However, an increase in oxidants or a decrease in antioxidants can disrupt this balance, leading to elevated ROS levels and potential oxidative stress.

Figure 4 illustrates the normalized levels of SOD when exposed to a 45 μT SMF alone and in combination with a 20 nT RF field across different frequencies. At 4.5 MHz, the treated group showed the highest increase in SOD levels (*p* < 0.001). These findings suggest that SOD levels are significantly influenced by the combined exposure to SMF and RF fields, particularly at certain frequencies.

Cancer cells rely heavily on antioxidant defense mechanisms to manage elevated levels of ROS, which are byproducts of increased metabolic activity. According to DeBerardinis et al., cancer cells increase their antioxidant capacity to avoid toxic ROS levels and maintain redox homeostasis, which is critical for tumor progression and metastasis. Elevated antioxidant enzyme activity, such as that of SOD, plays a key role in protecting cells from oxidative stress [47].

ROS are primarily produced at complex I and complex III of the mitochondrial ETC and are dissipated in mitochondria by antioxidants like SOD, GPx, and catalase, preventing cellular damage [48]. As organisms age, the efficiency of these antioxidant defenses diminishes, leading to an accumulation of oxidative damage, which is a key contributor to the development of age-related diseases like neurodegenerative disorders and cancer [49]. The decline in mitochondrial function and the resulting increase in ROS production further exacerbate this process, creating a vicious cycle of oxidative stress and cellular dysfunction [50].

### 3.2. Changes in Oxidative Stress Markers: GSH (Reduced Glutathione)

Glutathione (GSH), the smallest intracellular thiol, protects cells from ROS-induced damage, including from free radicals and peroxides. Glutathione exists in two states: reduced (GSH) and oxidized (GSSG). GSH acts as a major tissue antioxidant by providing reducing equivalents for the glutathione peroxidase (GPx)-catalyzed reduction of lipid hydroperoxides to alcohols and H_2_O_2_ to water.

During this reduction process, a disulfide bond forms between two GSH molecules, resulting in the generation of oxidized glutathione (GSSG). Glutathione reductase (GR) then recycles GSSG back to GSH with the simultaneous oxidation of β-nicotinamide adenine dinucleotide phosphate (β-NADPH2). In healthy cells, more than 90% of the total glutathione pool is in the reduced form (GSH). However, when cells are exposed to elevated levels of oxidative stress, GSSG accumulates, leading to an increased GSSG-to-GSH ratio, which serves as an indicator of oxidative stress.

The normalized levels of GSH in response to exposure to a 45 μT SMF alone and in combination with a 20 nT RF field across different frequencies are shown in Figure 5. Significant changes were observed in GSH levels at several frequencies. At 2.5 MHz, a significant reduction was noted (*p* < 0.001). Conversely, at 4 MHz, the treated groups exhibited substantial increases in GSH levels compared to the control groups (*p* < 0.001). These results indicate a frequency-dependent response in GSH levels to combined SMF and RF field exposure.

### 3.3. H_2_O_2_ Activity: Peroxidase

The intensity of the fluorescence through the use of the peroxidase assay directly correlates with the amount of H_2_O_2_ present in the sample, providing a quantitative measure of H_2_O_2_ levels. Thus, by introducing HRP to cellular extracts, we can indirectly determine the concentration of H_2_O_2_ in the biological samples. In biological systems, H_2_O_2_ is a ROS generated as a byproduct of various metabolic processes, particularly within the mitochondria during cellular respiration. Peroxidases, including HRP used in the assay, are enzymes that catalyze the reduction of H_2_O_2_ to water. This reaction is vital in mitigating oxidative stress by removing excess H_2_O_2_, thereby preventing potential cellular damage.

Figure 6 presents the normalized peroxidase levels under the influence of a 45 μT SMF alone and combined with a 20 nT RF field. At 2 MHz, 4.5 MHz, and 5 MHz the control group showed significantly higher peroxidase levels compared to the treated group (*p* < 0.001). The peroxidase levels generally exhibited less variation compared to GSH levels, suggesting a more stable response to magnetic field exposure.

### 3.4. A Key ROS: Superoxide

ROS are not only implicated in cancer development but also in the aging process and age-related diseases [51]. Moreover, the increased ROS generation in aged cells exacerbates oxidative damage, which is a common hallmark of both aging and cancer, suggesting a shared mechanism underlying these conditions [52]. The mitochondrial production of ROS is particularly significant, as it contributes to the progressive decline in cellular function associated with aging [53].

Mitochondrial O_2_•^−^ is a critical ROS generated primarily in the mitochondria as a byproduct of oxidative phosphorylation. Key sites of superoxide production include complexes I and III, as well as glycerol phosphate dehydrogenase, all of which contribute to ROS levels either within the matrix or intermembrane space [54]. This production is a significant factor in mitochondrial oxidative stress, which is closely linked to aging and the progression of various diseases.

In our study, mitochondrial superoxide levels were assessed using the MitoSOX Red indicator, a fluorogenic dye selectively targeted to mitochondria, where it is rapidly oxidized by superoxide to exhibit red fluorescence. Cells exposed to various RF frequencies (ranging from 2 MHz to 5 MHz) under an SMF of 45 μT and a 20 nT RF field exhibited frequency-dependent changes in mitochondrial superoxide levels. As shown in Figure 7, mitochondrial superoxide levels were significantly reduced at 4.5 MHz compared to the control group (*p* < 0.001). Conversely, at 2 MHz, a significant increase in superoxide production was observed (*p* < 0.001). At 3.5 and 5 MHz, the superoxide levels in the treated group were slightly elevated compared to the control, but the effect was not as pronounced as at 2 MHz. These findings suggest that hyperfine resonance effects may differentially modulate mitochondrial superoxide production depending on the applied RF frequency. This frequency-specific modulation aligns with the hypothesis that weak RF fields influence mitochondrial ETC components through hyperfine interactions, altering superoxide generation.

Extracellular superoxide plays a crucial role in oxidative stress. It can result from the release or transport of intracellular superoxide (or related ROS) to the extracellular environment, often through mechanisms such as SOD enzymes or channels like anion transporters. To assess the impact of RF field exposure on oxidative stress in HT-1080 cells, extracellular superoxide levels were measured using EPR and the probe cyclic hydroxylamine (CMH). CMH reacts with superoxide to form a stable nitroxide radical (CM●), detectable by EPR. Cells with and without RF exposure were treated in KHB containing 100 µM DTPA and CMH (0.3 mM) and incubated for 50 min at 37 °C, with or without pretreatment with SOD1 (30 U/mL). After incubation, the supernatant was flash-frozen in Teflon tubing for EPR analysis. The spectra represent typical nitroxide spectra at 77 K. Figure 8 demonstrates the quantitative analysis of extracellular superoxide levels, showing a reduction in superoxide levels following exposure to RF fields at 4 MHz.

### 3.5. Cell Viability and Confluence Analysis

Cell viability, a crucial parameter in understanding the cellular response to external stimuli, particularly in cancer cells, is often influenced by oxidative stress and the balance between ROS and antioxidant defenses. This imbalance between ROS and antioxidants can promote tumor progression, as oxidative stress triggers H_2_O_2_-mediated signaling pathways that influence proliferation and survival [55]. Exposure to RF fields has been shown to modulate oxidative stress pathways, which in turn affects cell survival. In particular, changes in ROS levels, such as O_2_•^−^ and H_2_O_2_, can lead to either protective antioxidant responses or cell damage, depending on the frequency and intensity of exposure.

Cell viability was evaluated using the CellTiter 96 Non-Radioactive Cell Proliferation Assay. This assay measures the metabolic activity of cells as an indicator of viability, wherein viable cells convert a tetrazolium compound into a colored formazan product. As shown in Figure 9, cells exposed to RF fields at 4 MHz exhibited significantly increased viability (*p* < 0.01), corresponding with the enhanced antioxidant response and reduced oxidative stress markers observed at this frequency. This supports the hypothesis that optimal resonance effects at 4 MHz promote cell survival by modulating mitochondrial function and reducing ROS-induced damage. In contrast, at 3.5 MHz, cell viability was markedly reduced (*p* < 0.01), correlating with the heightened oxidative stress and GSH depletion observed at this frequency, suggesting that excessive ROS production may lead to increased cell death.

HT-1080 cell images were captured, and confluence analysis was conducted using the Celigo Image Cytometer (Nexcelom Bioscience, Lawrence, MA, USA). The confluency percentages were calculated for both treated and control groups. In Figure 10, images were taken from representative samples that closely reflected the average confluency observed across replicates. The experiment was repeated twice with eight samples each (*n* = 16, *N* = 2). The control group exhibited an average confluency of 60.25% ± 9.01, while the treated group showed an average confluency of 73.12% ± 7.93. A significant difference was observed between the control and treated groups, with a *p*-value of 0.00017. We specifically chose to perform the confluence analysis at 4 MHz due to its notable impact on cell viability and antioxidant responses. This frequency exhibited significant biological effects, including enhanced cell proliferation and optimized mitochondrial function, making it a key point for further investigation.

### 3.6. Summary

Figure 11 illustrates the normalized responses of key biological markers, including SOD, peroxidase, GSH, mitochondrial superoxide, and cell viability in HT-1080 cells exposed to combined SMF and RF fields. The frequency range spans 2 MHz to 5 MHz, with a control line representing cells exposed to SMF alone.

## 4. Discussion

In biological systems, hyperfine resonance can influence enzyme activity if the enzyme contains paramagnetic centers such as metal ions. SOD, especially the forms containing metal ions (Cu/Zn-SOD and Mn-SOD), can be affected by hyperfine interactions because these metal centers are paramagnetic. Hyperfine interactions can modulate SOD activity, potentially enhancing the production of superoxide radicals or altering the enzyme’s ability to convert superoxide radicals into H_2_O_2_.

The differential responses of GSH and SOD at 2 MHz and 2.5 MHz can be attributed to hyperfine resonance effects on SOD. At 2.5 MHz, the increased SOD activity may be due to upregulation of SOD expression in response to the RF treatment, leading to significant oxidative stress management and GSH depletion. At 2 MHz, the effect is less pronounced, resulting in significant GSH depletion without a corresponding increase in SOD activity. This indicates a frequency-specific modulation of oxidative stress markers. Significant increases in GSH levels at 3 MHz and 4 MHz suggest enhanced mitochondrial efficiency and antioxidant responses.

Our previous data [56] showed significant cell growth around 4 MHz, aligning with the current findings of increased SOD and GSH levels and decreased H_2_O_2_ levels at this frequency. This suggests a potential link to hyperfine resonance effects on mitochondrial metalloproteins, leading to enhanced mitochondrial proliferation. This supports the hypothesis of robust antioxidant response and efficient handling of ROS at specific frequencies. In cancer cells, the balance between ROS generation and antioxidant responses is particularly finely tuned. Small increases in mitochondrial ROS can promote tumorigenesis, while excessive ROS can lead to cell death, underscoring the delicate equilibrium that these cells must maintain [57,58]. Phaniendra et al. [59] emphasize how the accumulation of ROS over time accelerates aging and increases susceptibility to various diseases by inducing damage to lipids, proteins, and DNA.

The findings from Kıvrak et al. provide substantial evidence that EMF exposure disrupts the antioxidant defense system, leading to increased oxidative stress [60]. Similarly, Snezhkina et al. [61] suggest that ROS generation in normal and malignant cells is primarily driven by mitochondrial electron leakage. Our study shows that RF fields can modulate mitochondrial ROS through hyperfine interactions affecting key paramagnetic centers like iron-sulfur clusters, altering electron transfer efficiency. The frequency-specific changes in SOD and GSH further support the role of these interactions in modulating ROS production and antioxidant enzyme activities. Furthermore, Zhang et al. demonstrated that shielding the geomagnetic field (HMF) significantly reduced H_2_O_2_ production in human neuroblastoma cells and inhibited the activity of CuZn-SOD [62]. This suggests that environmental magnetic fields, including those altered by RF exposure, can significantly impact ROS production and the activity of key antioxidant enzymes.

Superoxide measurements across varying frequencies revealed distinct patterns in mitochondrial ROS production. At both 4 MHz and 4.5 MHz, SOD activity significantly increased, while mitochondrial superoxide levels decreased. This suggests that RF exposure at these frequencies enhances the antioxidant defense by increasing SOD activity, thereby reducing mitochondrial superoxide levels. These findings indicate that 4 and 4.5 MHz may induce a resonance effect that optimizes the balance between ROS production and antioxidant response, supporting improved mitochondrial function. Cell viability assessments further confirm the relationship between mitochondrial function and RF exposure. The increased cell viability observed at 4 MHz corresponds with enhanced antioxidant responses and reduced oxidative stress markers, underscoring the role of specific frequencies in promoting cancer cell survival. Conversely, at 3.5 MHz, reduced viability and GSH depletion reflect heightened oxidative stress and potential cell damage. These findings emphasize the nuanced impact of RF frequency on ROS dynamics, cell survival, and mitochondrial function.

Biological responses to EMFs depend on parameters such as frequency, intensity, modulation, and exposure duration. Franco-Obregón emphasized the importance of experimental conditions, including ambient magnetic fields, temperature, and cell density, which can significantly impact the reproducibility and interpretation of results in EMF studies [63]. These factors need to be meticulously controlled to ensure consistent and reliable outcomes. Similarly, Portelli et al. [64] highlighted that background magnetic field variability in biological incubators can be a confounding factor, influencing experimental reproducibility. By addressing these variables, the effects observed in this study are likely attributable to the applied EMFs rather than environmental noise.

Our findings align with descriptions of quantum biological processes where weak magnetic fields influence the recombination rates of radical pairs. The RPM explains how weak magnetic fields can influence chemical reactions in biological systems by affecting the recombination rates of radical pairs [65]. According to Challis, the energy of a photon of an RF field is significantly lower than that required to ionize a typical molecule, making it unlikely for RF fields to cause direct ionization or excitation [66]. However, RF fields can influence radical pairs, which are intermediates in many chemical reactions and are formed with either antiparallel or parallel spins. The application of RF radiation at the hyperfine splitting frequency can increase the time these pairs spend in the triplet state, thus increasing the probability of their dissociation into free radicals. As shown by Ritz et al., even weak oscillating magnetic fields within the 0.1–10 MHz range at strengths as low as 85 nT can have significant biological effects, supporting the idea that radical pairs are sensitive to such fields [67].

Rodgers and Hore describe the RPM as involving pairs of transient radicals whose chemical fates are controlled by weak magnetic interactions via their spin states [68]. These radicals are created simultaneously and can exist in singlet or triplet states, which interconvert under the influence of magnetic fields. The singlet-triplet interconversion is modulated by hyperfine interactions and the Zeeman effect, which can alter the reaction yields depending on the orientation and strength of the external magnetic field. Timmel et al. demonstrated that even weak magnetic fields can significantly alter product yields, with free radical production being boosted and recombination suppressed by 10–40% depending on the resonance conditions. Any resonance effects from oscillating magnetic fields are expected to occur at frequencies between 1 and 100 MHz, matching the energy level splittings produced by hyperfine interactions in the radical pair [69]. The calculated Zeeman energies highlight the potential for RF fields to influence radical pairs by inducing transitions at specific frequencies. For example, the transition frequencies for electron spin states in the Earth’s magnetic field (~45 µT) typically fall between 1 and 10 MHz [70]. These transitions can alter the recombination rates of radical pairs, affecting the production of ROS and the activities of antioxidant enzymes.

Under the applied magnetic field intensities of 20 nT, we measured the resulting electric fields in the medium, revealing values of 1.128 V/m. Our study underscores the necessity for revising current exposure standards for RF fields, considering the observed biological effects at levels below existing safety guidelines. The frequency-specific modulation of oxidative stress markers and antioxidant responses emphasizes the need for updated guidelines to ensure public health safety. Currently, the International Commission on Non-Ionizing Radiation Protection (ICNIRP) sets the reference level for occupational exposure to RF fields in the 2–5 MHz region at 61 V/m. Given our findings, these standards may require reevaluation to better protect against potential biological effects.

Quantum biological principles such as coherence, entanglement, and resonance are fundamental to understanding these effects. Coherence refers to the property of electron spins maintaining a correlated state over time, which is crucial for the magnetic field to have a significant effect. Entanglement involves the interconnected state of electron spins, meaning the state of one electron can instantly influence the state of another. Resonance occurs when the frequency of the applied magnetic field matches the natural frequency of the radical pair system, leading to maximum sensitivity and effect. These principles are key to explaining how even very weak RF fields can produce significant biological effects. In the context of our study, applying RF fields at specific frequencies could influence the spin states of radical pairs formed during metabolic processes in cancer cells. Nunn et al. describe how quantum effects, such as tunneling and coherence, are vital for mitochondrial function, particularly in electron transport, which is essential for ATP production [71]. Disruptions in these processes, especially with aging, contribute to mitochondrial dysfunction and oxidative stress. Hormetic stress, such as that induced by weak RF fields, could enhance mitochondrial quantum efficiency, thereby improving redox balance and mitigating age-related mitochondrial decline.

While the results provide valuable understanding, some limitations must be addressed. The use of only one cancer cell line (HT-1080 human fibrosarcoma) may constrain the applicability of the findings to different cancer types or normal cells. Moreover, although multiple technical replicates were performed, only two independent experiments were conducted per parameter, potentially affecting the robustness of the conclusions. As this is an in vitro study, the results may not fully capture the complexity of in vivo environments. Future research should involve in vivo or in situ studies to validate these findings in more biologically relevant systems, explore a broader range of cell lines, increase the number of independent experiments, and confirm the frequency-specific effects on oxidative stress. Furthermore, repeating the experiment at different RF amplitudes would help clarify how amplitude variations influence the rates of response, particularly near the observed peak changes.

## 5. Conclusions

Our findings reveal that RF exposure at specific frequencies distinctly modulates the activities of key antioxidant enzymes, such as SOD and peroxidase, as well as GSH levels. This modulation underscores the dual role of SODs in controlling ROS damage and regulating ROS signaling, as detailed by Wang et al. [72]. By balancing ROS production and detoxification, SODs are crucial in maintaining cellular homeostasis and influencing cancer cell behavior. At 2.5 MHz, we observed significant oxidative stress, indicated by increased SOD activity and substantial GSH depletion, suggesting that hyperfine resonance effects at this frequency may amplify superoxide radical production, requiring a heightened antioxidant response. In contrast, exposure at 4 MHz led to a robust antioxidant response, characterized by elevated SOD and GSH levels and decreased H_2_O_2_. This coincided with an increase in cell viability and decreased mitochondrial superoxide, indicating enhanced mitochondrial function and an optimized balance between ROS production and antioxidant defenses in cancer cells. These frequency-dependent effects imply the involvement of hyperfine resonance interactions with mitochondrial metalloproteins, significantly impacting ROS dynamics. This modulation of ROS and apoptosis mirrors findings where RF-EMF exposure altered NADPH homeostasis and reduced superoxide levels, contributing to cell survival [73]. The potential for antioxidants to mitigate RF-induced oxidative stress underscores the need for further research into antioxidant therapies.

At an applied RF field intensity of 20 nT, our conclusions align with a broader body of evidence suggesting that low-intensity electromagnetic fields exert biological effects through non-thermal mechanisms [74]. Furthermore, concerns have been raised about the impact of RF field intensities below current safety standards, particularly for sensitive populations. While current exposure standards are primarily based on avoiding thermal effects, increasing evidence indicates that non-thermal influences of RF fields can modulate oxidative stress, ROS production, and other cellular processes [75]. Such findings call attention to the importance of re-evaluating these standards.

Investigating the therapeutic potential of combining RF exposure with antioxidant or pro-oxidant treatments could advance our understanding of how RF fields modulate cellular redox states. These findings suggest that frequency-specific RF exposure may selectively influence cancer cell oxidative stress, potentially improving the efficacy of conventional cancer therapies. Certain RF frequencies can either increase or decrease superoxide and other oxidative stress markers, thereby altering the balance between ROS production and antioxidant defenses. By increasing the susceptibility of cancer cells to oxidative damage, RF exposure could make them more vulnerable to treatments like chemotherapy or radiation, which depend on inducing oxidative stress. These mechanisms may also be relevant to age-related diseases, as oxidative stress is a shared factor in both cancer and aging, where mitochondrial dysfunction and ROS overproduction contribute to cellular decline, as noted by Hajam et al. [76]. Furthermore, the concept of mitohormesis, as described by Ristow [77], suggests that controlled ROS levels can trigger adaptive responses that protect cells from damage, thereby supporting longevity and healthy aging.

## Figures and Tables

**Figure 1 antioxidants-13-01237-f001:**
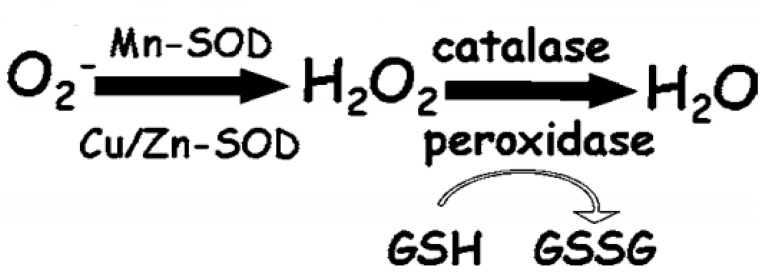
Interconnected activities of antioxidant enzymes like SOD, catalase, and peroxidase, and how they mitigate oxidative stress. Adapted from [35].

**Figure 2 antioxidants-13-01237-f002:**
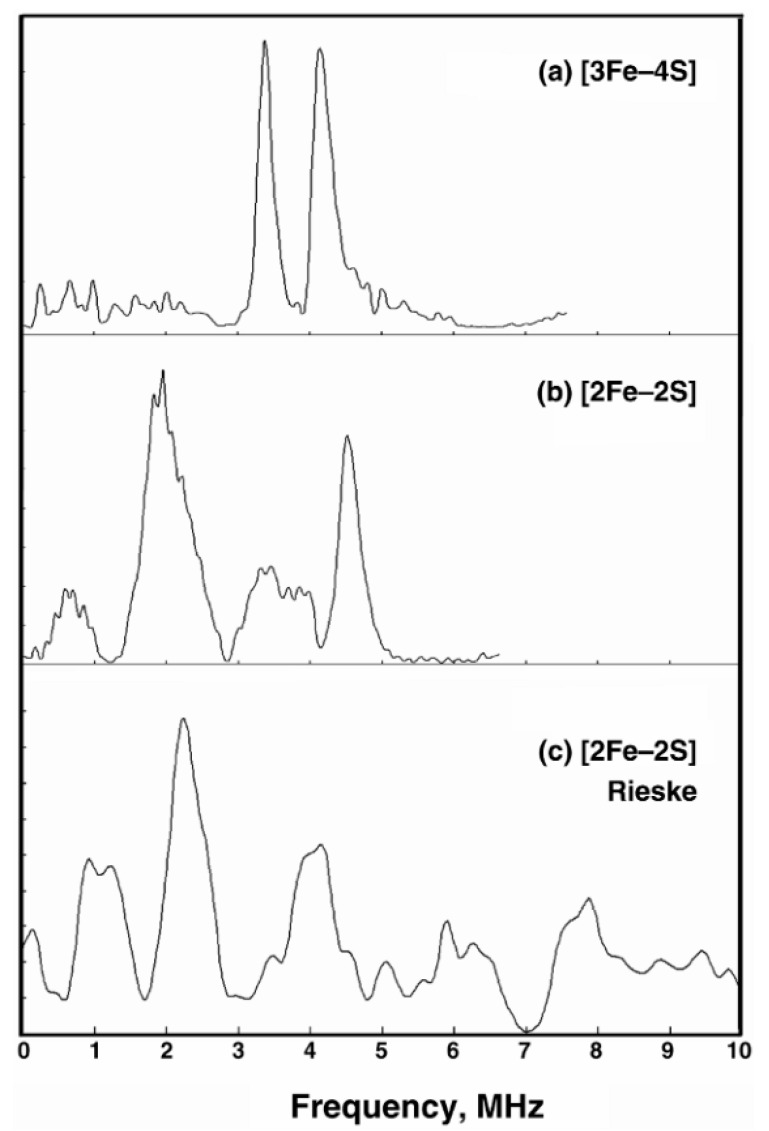
X-band ESEEM spectroscopy illustrates (**a**) the [3Fe–4S] cluster in fumarate reductase from *E. coli*; (**b**) the [2Fe–2S] ferredoxin from *A. platensis*; (**c**) the Rieske protein in bovine heart mitochondria. Adapted from [12].

**Figure 3 antioxidants-13-01237-f003:**
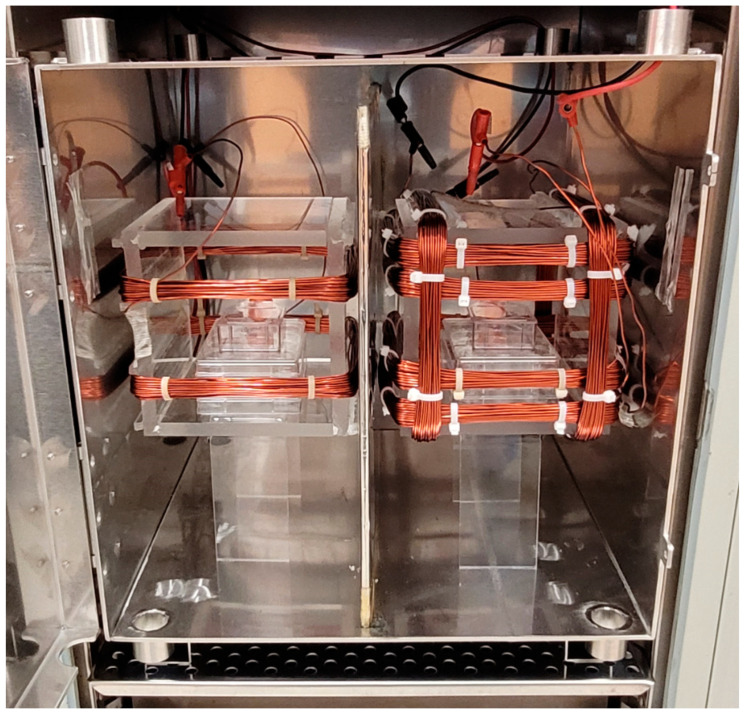
The experimental setup features two Helmholtz coils placed within a mu-metal cage. The left coil, with a single pair of turns, generates a SMF for the control cells. In contrast, the right coil consists of an inner coil that produces SMF, and an outer coil designed to generate an RF magnetic field for the treated cells. A mu-metal sheet separates the two coils within the cage, ensuring proper field isolation.

**Figure 4 antioxidants-13-01237-f004:**
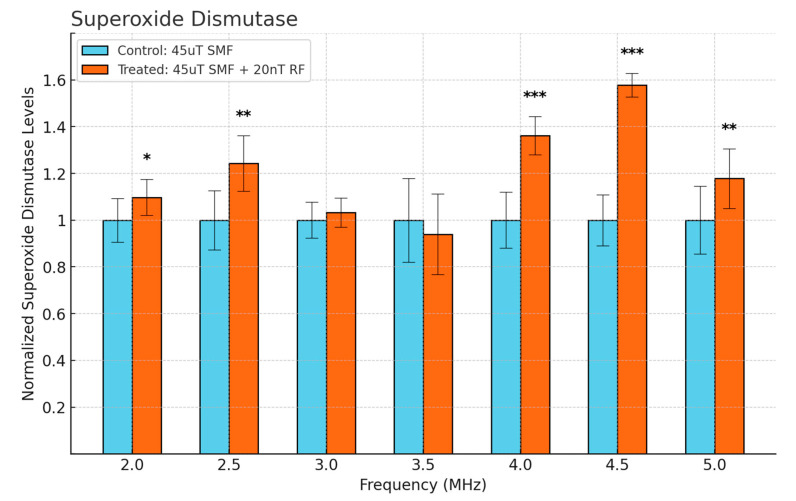
Cellular SOD levels as a function of frequency in fibrosarcoma cells. Data are expressed as mean ± SD (*n* = 18, *N* = 2) for each group. * *p* < 0.05, ** *p* < 0.01, and *** *p* < 0.001 represent significant differences.

**Figure 5 antioxidants-13-01237-f005:**
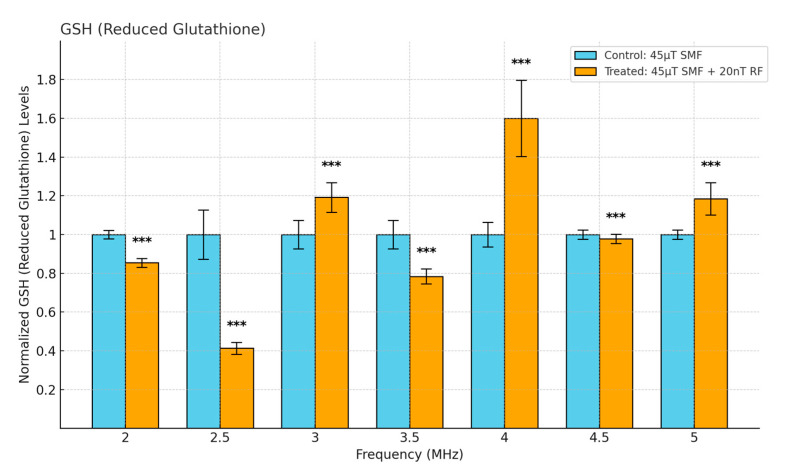
Cellular GSH levels as a function of frequency in fibrosarcoma cells. Data are expressed as mean ± SD (*n* = 50, *N* = 2) for each group. *** *p* < 0.001 represents significant difference.

**Figure 6 antioxidants-13-01237-f006:**
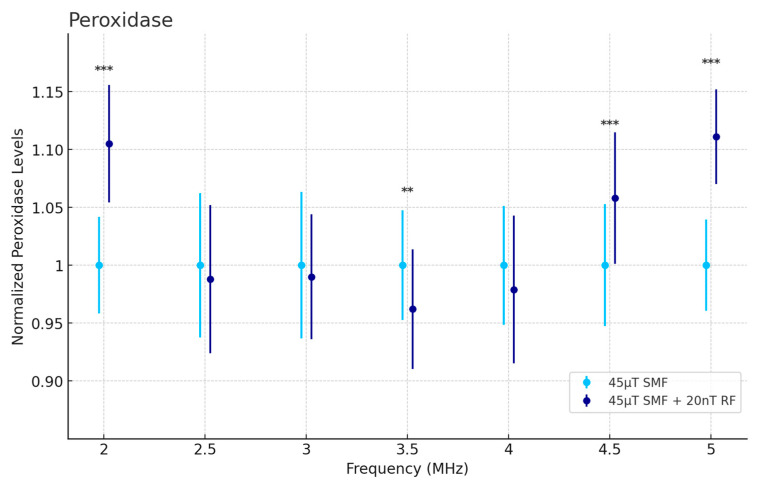
Peroxidase activity assay assessing cellular H_2_O_2_ levels as a function of frequency in fibrosarcoma cells. Data are expressed as mean ± SD (*n* = 36, *N* = 2) for each group. ** *p* < 0.01, and *** *p* < 0.001 represent significant differences.

**Figure 7 antioxidants-13-01237-f007:**
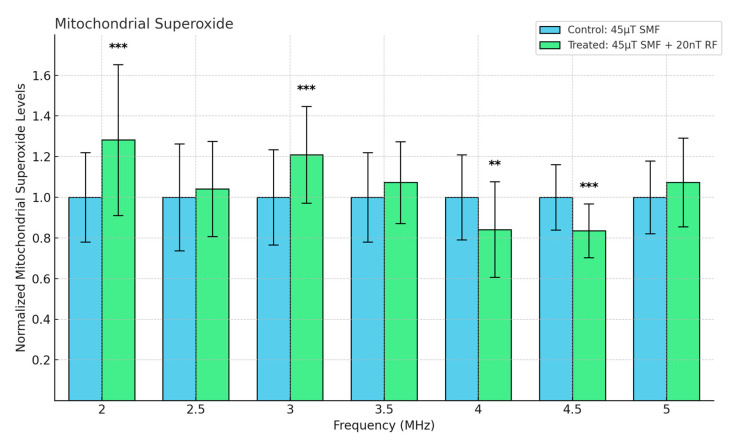
Mitochondrial Superoxide levels as a function of frequency in fibrosarcoma cells. Data are expressed as mean ± SD (*n* = 32, *N* = 2) for each group. ** *p* < 0.01, and *** *p* < 0.001 represent significant differences.

**Figure 8 antioxidants-13-01237-f008:**
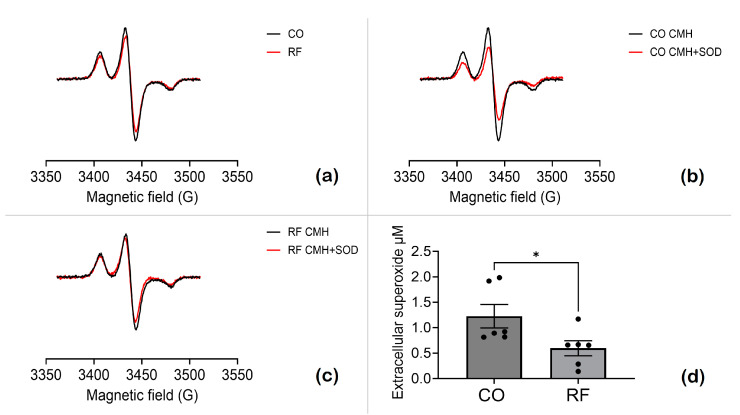
Extracellular superoxide levels in HT-1080 cells exposed to magnetic fields. CO refers to cells exposed to a 45 µT SMF, while RF indicates cells exposed to a 45 µT SMF combined with a 20 nT RF field at 4 MHz. (**a**) EPR spectra of CO cells (black trace) and RF-exposed cells (red trace). (**b**) EPR spectra of CO cells with (red) or without (black) pretreatment with SOD1. (**c**) EPR spectra of RF-exposed cells, with (red) or without (black) pretreatment with SOD1. The CM● concentration was determined by double integration, followed by analysis using SpinCount. (**d**) SOD-inhibitable signal reflecting extracellular superoxide levels. Data are expressed as mean ± SEM; * *p* < 0.05 represents significant difference.

**Figure 9 antioxidants-13-01237-f009:**
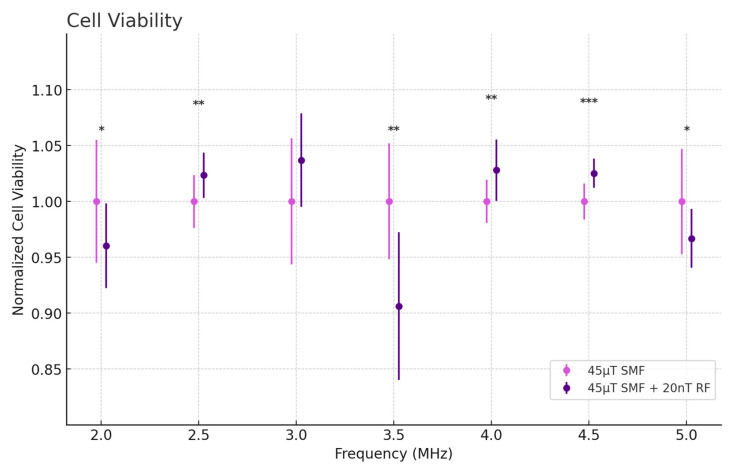
Cell Viability levels as a function of frequency in fibrosarcoma cells. Data are expressed as mean ± SD (*n* = 28, *N* = 2) for each group. * *p* < 0.05, ** *p* < 0.01, and *** *p* < 0.001 represent significant differences.

**Figure 10 antioxidants-13-01237-f010:**
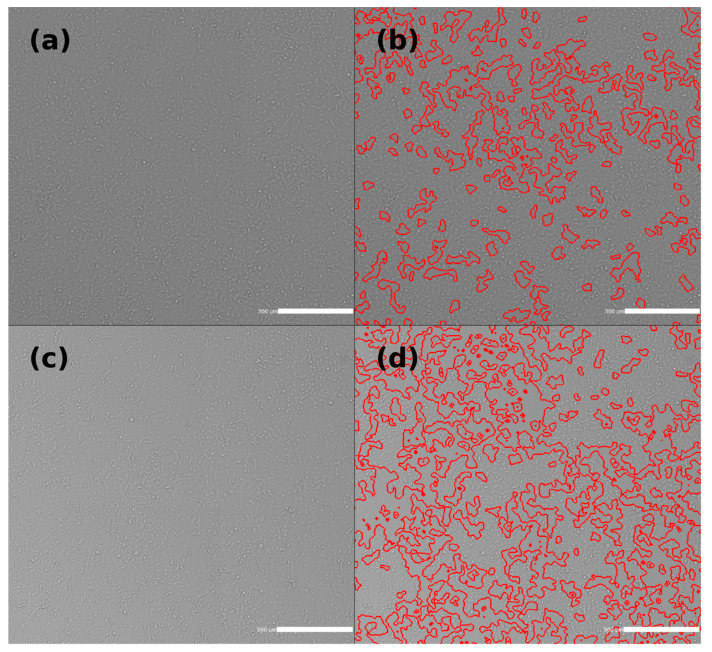
(**a**) HT-1080 cells exposed to a 45 µT SMF and 4 MHz RF field for 4 days (treated). (**b**) Confluence analysis of the cells shown in (**a**), with red contours outlining areas of low or no cell coverage. The confluency of this representative sample was calculated to be 72.77%. (**c**) HT-1080 cells exposed to a 45 µT SMF for 4 days (control). (**d**) Confluence analysis of the cells shown in (**c**), with red contours marking regions of low or no cell coverage. The confluency of this representative sample was calculated to be 60.05%.

**Figure 11 antioxidants-13-01237-f011:**
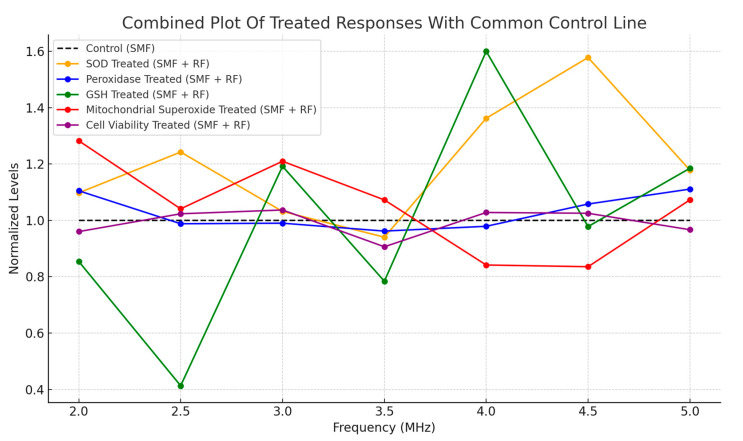
Combined plot of normalized biological responses to SMF and RF exposures across 2–5 MHz. Responses include SOD, GSH, mitochondrial superoxide, and cell viability, with the control condition exposed to SMF only.

## Data Availability

The data presented in this study are available on request from the corresponding author.

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
