# Peer review of "Frequency-Dependent Antioxidant Responses in HT-1080 Human Fibrosarcoma Cells Exposed to Weak Radio Frequency Fields"

_antioxidants, 2024, doi:10.3390/antiox13101237_

Round 1

Reviewer 1 Report

1The manuscript entitled "Understanding Radio Frequency Influence on Cancer Cells and Antioxidant Responses" focuses on the effects of RF on antioxidant factors in cancer cells. Despite being interesting, this study only deals with 3 parameters: reduced GSH, SOD activity and Peroxidase activity. The described results should be completed by other experiments (RNAseq, ROS measurement…). Of note, this paper is well written, with a quite comprehensive introduction.

11)      Figures 4, 5, 6: Only 2 independent replicates were performed. Please add a third replicate for all experiments.

22) The amount of data presented in this Ms is too low to be published as a regular article. The experiments on GSH/SOD/Peroxidase should be completed by other experiments to highlight the antioxidant capacity of cancer cells. RNAseq should be performed on treated cells, Mitosox could be used to measured mitochondrial superoxides, Catalase activity could be explored…

33)  Only 1 cancer cell line was used for these experiments. The data should be reproduced on another cancer cell line at least.

44) Actually, these data should be more precisely discussed in the light of the Authors previous results (PMID 37648766 & 37509147). A correlation with ROS levels in treated cells is necessary, as well as with mitochondrial mass and cell proliferation.

The Authors should also add as a final figure a schematic summary of their findings between 2 and 5 MHz in response to RF (GSH, GPX, SOD, Peroxides/peroxinitrites, pH, mito mass, proliferation, oxidative stress). It seems to be necessary to conclude on the different effects of RF for each particular frequency.  

5) Discussion : p13 L518: no peroxidase decrease could be observed at 4 MHz, only at 3,5 MHz (with an increase at 4.5 MHz). Please amend.

Minor points:

11)      In Introduction: please precise p4 L165 the objective(s) of this study: either to understand RF effects on cells contributing to oncogenesis induction/promotion, or to sensitize, via RF-mediated effects, cancer cells to specific therapies (or both?). For the 2d objective, maybe discuss the effects of RF on normal cells in order to evaluate the off-side effects.

22)      In Figure 1 : is “peroxidase” used for GPX? Maybe use GPX instead?

33)      Introduction p2 L85. Complex II is also an important source of ROS. Please add a reference

44)      M&M p7 l.279-280: Maybe authors could clarify this part. Indeed, they seem to use SMF as a control to “minimize external influences”. How did the Authors choose the particular setting for this SMF exposure (45 µT)?

55)      Why this specific 2-5 MHz frequency? Please clarify in M&M p7 l.268

66)      M&M p 7 l.281-289. This is not M&M. This should be discussed in the Discussion.

Reviewer 2 Report

Specific comments to the authors

The authors Hakki Gurhan and Frank Barnes of the submitted manuscript entitled "Understanding Radio Frequency Influence on Cancer Cells and Antioxidant Responses" experimentally investigated the possible interaction between radio frequency (RF) exposure and cancer cells on oxidative stress levels.

In summary, based on their in vitro investigations, the authors were able to demonstrate the potential role of antioxidants in mitigating the adverse effects of RF on cancer cells. Therefore, the authors concluded that the findings could provide valuable information for future therapeutic interventions and preventive strategies.

Overall, the manuscript provides some interesting information on the possible interaction of RF on oxidative stress levels in the chosen in vitro experimental setting. The manuscript (including the presentation) is mostly understandable and convincing. The methods are mostly well described. Although the results and discussion are clearly presented, the authors (see specific comments) need to make some major changes to improve the manuscript. In conclusion, the data presented are partly interesting. After incorporating the specific comments mentioned (see below), the manuscript has the potential to be accepted.

Main concerns

The main concern is based on the use of only one human cancer cell line and the lack of a normal control cell line. In addition, histomorphological and immunohistochemical characterisation of the RF-treated cell line is completely lacking. Furthermore, the influence of oxidative stress-related drugs should be investigated by the authors. Finally, it is not clear what the differences are between the submitted manuscript and their own publications (such as "Sci Rep. 2023 Aug 30;13(1):14223. doi: 10.1038/s41598-023-41167-5" and "Biomolecules. 2023 Jul 13;13(7):1112. doi: 10.3390/biom13071112."), which should be clarified by the authors.

Minor concerns

Abstract: The summary of the results and conclusions is largely unspecific and should be more focused on the actual findings and final conclusions.

Introduction: The introduction is definitely too long and needs to be substantially shortened as the submitted manuscript is an article and not a review.

Materials and methods: It is not clear why the HT-1080 human fibrosarcoma cell line was chosen for the experiments. Please explain.

Results:

# Figure 1: Please indicate how RF is involved in the chemical process of the oxidative stress presented (see Chapter 1.3.).

# Figure 2: Please indicate the main changes in terms of chemical structures.

# Figure 4: Time-dependent analyses are missing. Normal control cell lines are missing.

# Figure 5: Time dependent analyses are missing. Normal control cell lines are missing.

# Figure 6: Time dependent analyses are missing. It is not clear why the presentation of Figure 6 is different from Figures 4 and 5. Normal control cell lines are missing.

The authors should compare the levels of oxidative stress between Figures 4, 5 and 6.

Discussion: What is the actual conclusion of the results? Please clarify. And how can the findings be used in therapeutic aspects? Please discuss briefly. Finally, the limitations of this in vitro study must be mentioned in detail (no in vitro, no in situ study, no drug modulations...).

Reviewer 3 Report

Journal: Antioxidants

Manuscript number#: 3094536

Article type: Research Paper

Title: Understanding Radio Frequency Influence on Cancer Cells and Antioxidant Responses

Decision: Reject

Comments:

Dear author(s),

The study attempts to deal with the critical issue of the influence of Radio Frequency on HT-1080 human fibrosarcoma cell line. Understanding the mechanisms of the effect of radio frequencies is essential to better the understanding of different waves and their effect on cancer cells. However, several notable points must be addressed before considering this work suitable for publication in a reputable journal:

·         The information provided in the introduction is very long and jargoning. It gives the expression of a narrative review paper. So, it is recommended that the introduction be rewritten, and the information provided in the literature review should be concise to at least half of the current version.

·         There is no need for figures 1 and 2 for the introduction section. These can be removed.

·         Only the HT-1080 human fibrosarcoma cell line was used in this study, so this study can’t be used as a reference to depict cancer cells as written in the title. The title of the Article should be revised to mention the cancer type.

·         The figure for the Exposure System (Figure 3) can be created in 2d/3d for better visualization of the system developed for this study. Use professional author services for such designs.

·         The manuscript is more focused on the Antioxidant Responses of the Radio Frequency on the HT-1080 human fibrosarcoma cell line and has no data regarding the influence of Radio Frequency on cancer cells, such as cell viability and cell morphology. You can argue that there are some studies, but different experimental conditions affect these results and should be included in this study. 

·         What were the other effects caused by these radio frequencies, and what was the mechanism behind the induction of such effects?

·         This study lacks a strong rationale for conducting the study. In the last paragraph of the introduction, the authors mentioned that “in some studies, increased ROS production and oxidative damage in RF-exposed cells, while others have found no significant effects.” The results are similar to previous studies without adding new information regarding why there was a discrepancy in previous studies. Moreover, this study modulates the activities of key antioxidant enzymes but lacks the underlying mechanisms.

·         This study lacks novelty and is based on previous studies.

Journal: Antioxidants

Manuscript number#: 3094536

Article type: Research Paper

Title: Understanding Radio Frequency Influence on Cancer Cells and Antioxidant Responses

Decision: Reject

Comments:

Dear author(s),

The study attempts to deal with the critical issue of the influence of Radio Frequency on HT-1080 human fibrosarcoma cell line. Understanding the mechanisms of the effect of radio frequencies is essential to better the understanding of different waves and their effect on cancer cells. However, several notable points must be addressed before considering this work suitable for publication in a reputable journal:

·         The information provided in the introduction is very long and jargoning. It gives the expression of a narrative review paper. So, it is recommended that the introduction be rewritten, and the information provided in the literature review should be concise to at least half of the current version.

·         There is no need for figures 1 and 2 for the introduction section. These can be removed.

·         Only the HT-1080 human fibrosarcoma cell line was used in this study, so this study can’t be used as a reference to depict cancer cells as written in the title. The title of the Article should be revised to mention the cancer type.

·         The figure for the Exposure System (Figure 3) can be created in 2d/3d for better visualization of the system developed for this study. Use professional author services for such designs.

·         The manuscript is more focused on the Antioxidant Responses of the Radio Frequency on the HT-1080 human fibrosarcoma cell line and has no data regarding the influence of Radio Frequency on cancer cells, such as cell viability and cell morphology. You can argue that there are some studies, but different experimental conditions affect these results and should be included in this study. 

·         What were the other effects caused by these radio frequencies, and what was the mechanism behind the induction of such effects?

·         This study lacks a strong rationale for conducting the study. In the last paragraph of the introduction, the authors mentioned that “in some studies, increased ROS production and oxidative damage in RF-exposed cells, while others have found no significant effects.” The results are similar to previous studies without adding new information regarding why there was a discrepancy in previous studies. Moreover, this study modulates the activities of key antioxidant enzymes but lacks the underlying mechanisms.

·         This study lacks novelty and is based on previous studies.

Round 2

Reviewer 1 Report

I would like to thank the Authors for their response to reviewer comments. 

The additional data and the modifications of requested points improve the quality of the Ms.

Several minor updates have still to be performed.

1) Figure 7: p14 L549: the Authors said "As shown in Figure 7, the highest superoxide production was observed at 4.5 MHz, where mitochondrial superoxide levels significantly increased compared to the control group (p < 0.001)". However, according to Fig7 and discussion p18 L664, it is quite the opposite. Please amend.

2) Figure 10 p16 : please give for this experiment a % of confluency  +/- SD for several replicates.

3) Please add in the discussion these 2 limitations :

            a) one cell line used only

           b) only 2 independent experiments / figure (but an important number of technical replicates)

Reviewer 2 Report

Specific comments to the authors

In the revised version of the manuscript, the authors were able to address the previously mentioned concerns in a very adequate and convincing manner. Therefore, the revised manuscript "Frequency-Dependent Antioxidant Responses in HT-1080 Human Fibrosarcoma Cells Exposed to Weak Radio Frequencies" should be accepted.

See major comments.
